# DemaFormer: Damped Exponential Moving Average Transformer with Energy-Based Modeling for Temporal Language Grounding

**Thong Nguyen[1], Xiaobao Wu[2], Xinshuai Dong[3],**
**Cong-Duy Nguyen[2], See-Kiong Ng[1], Luu Anh Tuan[2]\***
[1]National University of Singapore, Singapore
[2]Nanyang Technological University, Singapore
[3]Carnegie Mellon University, USA
e0998147@u.nus.edu, anhtuan.luu@ntu.edu.sg

## Abstract

Temporal Language Grounding seeks to localize video moments that semantically correspond to a natural language query. Recent advances employ the attention mechanism to learn the relations between video moments and the text query. However, naive attention might not be able to appropriately capture such relations, resulting in ineffective distributions where target video moments are difficult to separate from the remaining ones. To resolve the issue, we propose an energy-based model framework to explicitly learn moment-query distributions. Moreover, we propose DemaFormer, a novel Transformer-based architecture that utilizes exponential moving average with a learnable damping factor to effectively encode moment-query inputs. Comprehensive experiments on four public temporal language grounding datasets showcase the superiority of our methods over the state-of-the-art baselines. Our code and data are publicly available at https://github.com/... (the link is hidden now due to the double-blind review).

## 1 Introduction

Temporal Language Grounding (TLG) is a task to determine temporal boundaries of video moments that semantically correspond (relevant) to a language query (Hendricks et al., 2018; Gao et al., 2021a). TLG is a complex and challenging task, since video processing demands understanding across multiple modalities, including image, text, and even audio. However, TLG has received increasing attention in CV and NLP communities because it provides myriad usage for further downstream tasks, e.g. VQA (Lei et al., 2018; Ye et al., 2017; Wang et al., 2019a), relation extraction (Gao et al., 2021b), and information retrieval (Ghosh et al., 2019).

Early methods for TLG deploy concatenation with linear projection (Gao et al., 2017; Wang et al.,

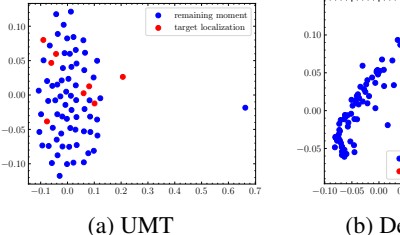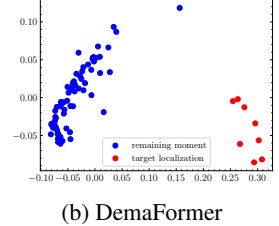

(a) UMT  (b) DemaFormer

Figure 1: Visualization (t-SNE) of moment-query representations of an input example by the previous best UMT baseline and our DemaFormer model. The target localizations are from the QVHighlights dataset label. Detailed input content is provided in Figure 2.

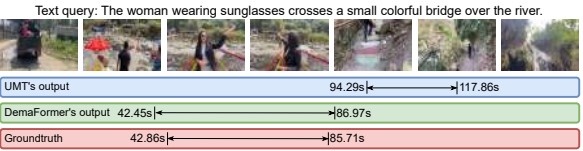

Figure 2: A TLG example. To produce the output, we form the union of overlapped temporal boundaries in the groundtruth and models' localized moments. The UMT output is about countryside scenes, which hardly align with the language query.

2019b) or similarity functions (Anne Hendricks et al., 2017; Hendricks et al., 2018) to fuse textual and visual features. To further enhance the localization performance, recent works divide the video into equal-length moments and employ the attention mechanism of the Transformer model to learn the relations between video moments and the language query. For example, Moment-DETR model (Lei et al., 2021) concatenates the visual moments and the textual tokens, and then passes the concatenated sequence to the Transformer encoder to capture the alignment. The UMT model (Liu et al., 2022) includes an additional audio channel into the Transformer encoder to construct a unified architecture.

However, for the TLG task, previous works have shown that such attention-based approach is still insufficient to capture the rich semantic interaction

---
*Corresponding Author

between the text query and video moments (Xu et al., 2023). As in Figure 2, the localized video moments hardly align with the query statement. Moreover, the attention mechanism in the Transformer encoder does not assume any prior knowledge towards the input elements (Ma et al., 2022). For language localization, this design choice does not leverage the fact that video moments in temporal neighborhoods tend to exhibit closely related features. Therefore, this approach could lead to ineffective modeling of joint moment-query inputs. As evidence, Figure 1 demonstrates the distribution of joint moment-query representations. Particularly, the representations of target moment-query localizations and the remaining ones mingle together, making the grounding task more challenging.

To address these limitations, we dive deeper into polishing the distribution of moment-query representations. In addition to supervised training to correctly localize the language in the video, we perform unsupervised training to explicitly maximize the likelihood of moment-query localizations. This could help the multimodal model focus on capturing the distribution of target moment-query pairs and distinguishing them from others.

As such, we propose to model the distribution of moment-query representations under the framework of the Energy-Based Model (EBM). In contrast to other probabilistic models such as normalizing flow (Rezende and Mohamed, 2015) or autoencoder (Kingma and Welling, 2013), the EBM framework allows us to directly integrate the video moment's salience score into the density function, which results in accurate modeling of moment-query representations. Our implementation develops into a contrastive divergence objective which aims to minimize the energy of the relevant localizations while maximizing the energy of the deviating ones. Accordingly, the framework needs negative samples to represent high-energy regions. Therefore, we adapt the Langevin dynamics equation to directly sample negative inputs from the EBM distribution. Such approach is appropriate because in the beginning the distribution will not match the true distribution, hence the generated samples are assured to be negative. As the training progresses, the distribution will approximate the true distribution, consequently the Langevin equation is able to produce hard negative samples.

In addition, we incorporate the inductive bias that captures local dependencies among the moment-query inputs. We propose DemaFormer in which we equip the **D**amped **E**xponential **M**oving **A**verage (DEMA) computation for the Trans**F**ormer architecture. Technically, the computation applies exponentially decaying factors that consider the information from adjacent inputs. We further introduce learnable damping coefficients to enable the model to absorb adjacent information in a sufficient manner that ensures distinction among inputs. Eventually, we combine the DEMA computation with the attention mechanism to construct DemaFormer encoder and decoder modules.

To sum up, the contributions of our paper can be summarized as follows:

- We propose DemaFormer, a novel architecture for temporal language grounding. DemaFormer integrates exponential moving average with learnable damping coefficients into the attention mechanism to appropriately capture dependency patterns among video-language inputs.

- We propose a novel energy-based learning framework for temporal language grounding. The objective for the energy-based model can be formulated as a contrastive divergence to assist a classical grounding loss for modeling moment-query representations.

- We conduct extensive experiments to demonstrate the superiority of our method over previous state-of-the-art baselines. Furthermore, we conduct comprehensive ablation studies to evaluate our component proposals and deliver meaningful insights.

## 2 Related Work

**Temporal Language Grounding (TLG).** Introduced by Gao et al. (2017); Anne Hendricks et al. (2017), TLG is to locate relevant video moments given a language query. Early approaches use LSTM to encode language query and CNN for visual clips, and then estimate cross-modal similarity scores (Anne Hendricks et al., 2017; Hendricks et al., 2018). Modern techniques leverage attention mechanism and structured graph network to learn the video-language relationship (Xiao et al., 2021; Gao et al., 2021a; Zhang et al., 2020a; Yuan et al., 2019). Recent works (Liu et al., 2022; Lei et al., 2021) apply Transformer components to eliminate hand-crafted pre-processing and post-processing steps and make the model end-to-end trainable.

**Vision-Language Representation Learning.**
Modeling the vision-language interaction is important for vision-language tasks (Gao et al., 2021a; Nguyen et al., 2022b,a, 2023; Wei et al., 2022, 2023, 2024). Previous works propose diverse techniques, including circular matrices (Wu and Han, 2018), dynamic filters (Zhang et al., 2019), Hadamard product (Zhang et al., 2020a), and contrastive learning (Nguyen and Luu, 2021). To better learn fine-grained token-level and moment-level cross-modal relations, several authors adapt graph neural networks with graph convolutional techniques (Gao et al., 2021a; Zhang et al., 2019).

## 3 Methodology

Our task is to localize moments in videos from natural language queries. Formally, given a language query $q$ of $L_q$ tokens and a video $v$ composed of $L_v$ equal-length input moments, where each moment is represented by a visual frame sampled from the moment, we aim to localize $L_m$ time spans from $v$ that is aligned with the query $q$, noted as $\{(l_i, r_i)\}_{i=1}^{L_m}$, where each moment spans from $l_i$ to $r_i$ scaled by the video timelength and $L_m < L_v$.

Thus, we first describe our proposed damping exponential moving average attention for modeling video-language inputs in Section 3.1, the overall architecture in Section 3.2, and the training strategy empowered with energy-based modeling in Section 3.3 and 3.4.

### 3.1 Damping Exponential Moving Average (DEMA) Attention

In this section, we consider the input to our DemaFormer encoder $X_e$ and decoder $X_d$ in Section 3.2 as the general input $X = \{\mathbf{x}_i\}_{i=1}^{L_X}$ of length $L_X$. We delineate the exponential moving average (EMA) with the damping influence applied on $X$ as follows.

**DEMA Computation.** At first, we use a linear layer to map each input $\mathbf{x}_i$ to an intermediate space:

$$\mathbf{g}_i = \text{Linear}(\mathbf{x}_i), \quad (1)$$

Then, we estimate the current hidden state $\mathbf{l}_i$ as the sum of the previous hidden state $\mathbf{l}_{i-1}$ and the current intermediate input $\mathbf{g}_i$ with the weighting coefficients that decrease exponentially and are relaxed by damping coefficients:

$$\mathbf{l}_i = \boldsymbol{\alpha} \odot \mathbf{g}_i + (1 - \boldsymbol{\alpha} \odot \boldsymbol{\delta}) \odot \mathbf{l}_{i-1}, \quad (2)$$

where $\boldsymbol{\alpha} \in (0, 1)^d$ denotes the weighting coefficients, $\boldsymbol{\delta} \in (0, 1)^d$ the damping coefficients, and $\odot$ the elementwise product. Both $\boldsymbol{\alpha}$ and $\boldsymbol{\delta}$ are randomly initialized and learnable during training. Subsequently, we project the hidden state $\mathbf{l}_i$ back to the original input space:

$$\mathbf{x}_i' = \text{Linear}(\mathbf{l}_i). \quad (3)$$

**DEMA Attention.** Given the input $X$, we obtain the DEMA output in Eq. (3) and pass the output through a non-linear layer:

$$X' = \text{DEMA}(X), \quad (4)$$
$$Z = \text{SiLU}(\text{Linear}(X')), \quad (5)$$

where SiLU denotes the self-gated activation function (Ramachandran et al., 2017; Elfwing et al., 2018). We experiment with other activation functions in Appendix D. Subsequently, we perform the attention operation and utilize $Z$ which exhibits local dependencies as the value tensor:

$$Q = \text{Linear}(X), \quad (6)$$
$$K = \text{Linear}(X), \quad (7)$$
$$V = \text{Linear}(Z), \quad (8)$$
$$Z' = \text{softmax}\left(\frac{QK^T}{\sqrt{d_K}}\right) \cdot V, \quad (9)$$

where $d_K$ denotes the dimension of $K$. Thereafter, we aggregate the original input $X$ and the attention output $Z'$ in an adaptive manner:

$$\boldsymbol{\lambda} = \text{sigmoid}(\text{Linear}(X')), \quad (10)$$
$$P = \text{SiLU}(\text{Linear}(X') + \text{Linear}(Z \odot Z')), \quad (11)$$
$$H = \boldsymbol{\lambda} \odot P + (1 - \boldsymbol{\lambda}) \odot X. \quad (12)$$

### 3.2 Overall Architecture

Figure 3 illustrates our DemaFormer model for temporal language grounding. We explain the architecture in details in the following.

**Uni-modal Encoding.** Given a video $v$ consisting of $L_v$ moments and a text query with $L_q$ tokens, we employ pre-trained models to extract visual moment features $F = \{\mathbf{f}_i\}_{i=1}^{L_v}$, textual features $T = \{\mathbf{t}_i\}_{i=1}^{L_q}$, and audio features $A = \{\mathbf{a}_i\}_{i=1}^{L_v}$.

**Audio-Dependent Video Encoding.** For video-audio encoding, because audio signals possess heavy noisy information (Liu et al., 2022; Badamdorj et al., 2021), we only perform 1-layer

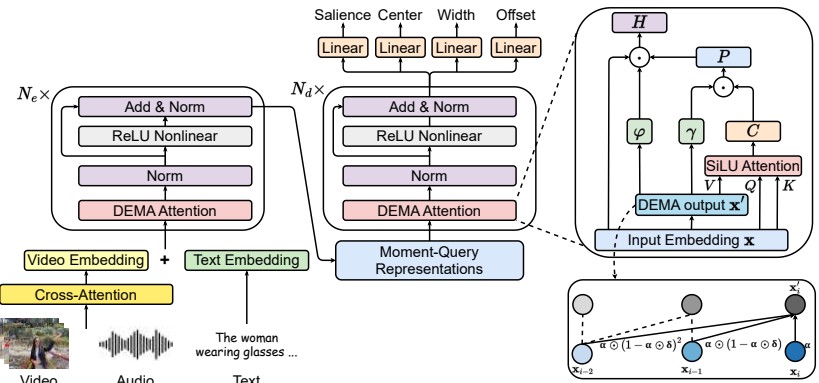

Figure 3: Illustration of the proposed DemaFormer. Our archtiecture comprises an encoder of $N_e$ layers and a decoder of $N_d$ layers. We designate the first $L_v$ encoder outputs as moment-query representations to become the input for the DemaFormer decoder.

attention to fuse the audio information into the visual sequence. Particularly, the attention between the video and audio input becomes:

$$F' = F + \text{softmax}\left(\frac{AF^T}{\sqrt{d}}\right) \cdot F. \quad (13)$$

**DemaFormer Encoder.** Inspired by (Lei et al., 2021), we concatenate the audio-dependent video and language tokens to form the input sequence:

$$X_e = [F', T]. \quad (14)$$

We push the input sequence $X_e$ to the DemaFormer encoder of $N_e$ encoder layers. Each encoder layer comprises a DEMA attention layer, a normalization layer, a ReLU non-linear layer, and a residual connection:

$$H_e^{(i+1)} = \text{Norm}\left(\text{DEMA}\left(X_e^{(i)}\right)\right), \quad (15)$$

$$X_e^{(i+1)} = O_e^{(i)} = \text{Norm}\left(\text{ReLU}\left(H_e^{(i)}\right) + H_e^{(i)}\right). \quad (16)$$

where $X^{(i)}$ and $O_e^{(i)}$ denote the input and the output of the $i$-th encoder layer, respectively; $H_e^{(i)}$ the intermediate output of the $i$-th encoder layer.

We take the output of the $N_e$-th encoder layer as the final output $O_e$ of the DemaFormer encoder.

$$O_e = O_e^{(N_e)}. \quad (17)$$

**DemaFormer Decoder.** The input for the decoder is the first $L_v$ DemaFormer encoder outputs, i.e. $X_d = \{\mathbf{o}_{e,i}\}_{i=1}^{L_v}$. The input sequence is forwarded to $N_d$ decoder layers, each of which is composed of a DEMA attention layer, a normalization layer,

a non-linear layer, and a residual connection:

$$H_d^{(i)} = \text{Norm}\left(\text{DEMA}\left(X_d^{(i)}\right)\right), \quad (18)$$

$$M^{(i+1)} = O_d^{(i)} = \text{Norm}\left(\text{ReLU}\left(H_d^{(i)}\right) + H_d^{(i)}\right). \quad (19)$$

Analogous to the encoder, we retrieve the $N_d$-th layer output as the final output $O_d$ of the decoder:

$$O_d = O_d^{(N_d)}. \quad (20)$$

**Prediction Heads.** For each output $\mathbf{o}_{d,i}$, we designate four separate linear layers to predict the salience score $\hat{s}_i$, the center $\hat{c}_i$, the center offset $\hat{co}_i$, and the moment width $\hat{w}_i$:

$$\hat{s}_i = \text{Linear}(\mathbf{o}_{d,i}), \ \hat{c}_i = \text{Linear}(\mathbf{o}_{d,i}), \quad (21)$$

$$\hat{co}_i = \text{Linear}(\mathbf{o}_{d,i}), \ \hat{w}_i = \text{Linear}(\mathbf{o}_{d,i}). \quad (22)$$

Thus, each candidate moment's temporal bound becomes $\left(\hat{c}_t + \hat{co}_i - \frac{\hat{w}_i}{2}, \hat{c}_i + \hat{co}_i + \frac{\hat{w}_i}{2}\right)$. At test time, we extract the top-$L_m$ moments whose salience scores are the largest.

### 3.3 Energy-Based Models for Modeling Moment-Query Representations

Given our joint video-language decoder outputs $O_d = \{\mathbf{o}_{d,i}\}_{t=1}^{L_v}$, we designate the EBM to specify the density of $O_d$ via the Boltzmann distribution:

$$p_\theta(\mathbf{o}_{d,i}) = \frac{\exp(-E_\theta(\mathbf{o}_{d,i}))}{Z_\theta}, \quad (23)$$

where $E_\theta$ denotes the energy function and $Z_\theta$ the normalizing constant $Z_\theta = \int \exp\left(-E_\theta(\mathbf{o}_{d,i})\right) \mathrm{d}\mathbf{o}_{d,i}$. Inspired by (Du and Mordatch, 2019), we adopt Langevin

dynamics to conduct sampling from the above distribution:

$$\tilde{\mathbf{o}}_{d,i}^{(k)} = \tilde{\mathbf{o}}_{d,i}^{(k-1)} - \frac{\gamma}{2}\nabla_{\mathbf{o}_{d,i}} E_\theta\left(\tilde{\mathbf{o}}_{d,i}^{(k-1)}\right) + \epsilon^{(k)},$$
(24)

$$\epsilon^{(k)} \sim \mathcal{N}(0,\gamma), \quad \tilde{\mathbf{o}}_{d,i} = \mathbf{o}_{d,i}^{(0)},$$
(25)

where $\gamma$ is a hyperparameter to specify the variance of the noise. We perform the Eq. (24) for $K$ steps and take $\tilde{\mathbf{o}}_{d,i}^{(K)}$ as the sampling outcome. Our target is to better align the video-query representation $O_d$, by minimizing the negative log likelihood of the moment-query representations, i.e. $\mathcal{L}_{\mathrm{NLL}}(\theta) = -\mathbb{E}_{\mathbf{o}_{d,i}}[\log p_\theta(\mathbf{o}_{d,i})]$. This can be achieved by differentiating the $\mathcal{L}_{\mathrm{NLL}}(\theta)$ and optimize the resulting contrastive divergence for the gradient, as:

$$\nabla_\theta \mathcal{L}_{\mathrm{NLL}} = \mathbb{E}_{\mathbf{o}_{d,i}^+}\left[\nabla_\theta E_\theta(\mathbf{o}_{d,i}^+)\right] - \mathbb{E}_{\mathbf{o}_{d,i}^-}\left[\nabla_\theta E_\theta(\mathbf{o}_{d,i}^-)\right],$$
(26)

whose detailed derivation can be found in Appendix A. Because the samples generated by Eq. (24) do not approximate the true distribution in the beginning but will gradually converge to, we take these samples as $\mathbf{o}_{d,i}^-$ and assign their energy values a decaying weight $\alpha$ with minimum value of $\alpha_{\min}$. We take the moment-query inputs whose groundtruth salience scores are larger than a threshold $\rho$ as positive samples $\mathbf{o}_{d,i}^+$.

Moreover, because we maximize salience scores while minimizing the energy values of the positive input (vice versa for the negative input), we implement the negative salience-energy relation:

$$E_\theta(\mathbf{o}_{d,i}) = -\hat{s}_i.$$
(27)

As such, $\theta$ becomes the DemaFormer's parameters and we obtain the final $\mathcal{L}_{\mathrm{NLL}}$'s formulation:

$$\alpha = \max\left(\frac{1}{1+\frac{1}{2}n_{\mathrm{epoch}}}, \alpha_{\min}\right),$$
(28)

$$\mathcal{L}_{\mathrm{NLL}} = \mathbb{E}_{\mathbf{o}_{d,i}^+}\left[E_\theta(\mathbf{o}_{d,i}^+)\right] - \alpha \cdot \mathbb{E}_{\mathbf{o}_{d,i}^-}\left[E_\theta(\mathbf{o}_{d,i}^-)\right],$$
(29)

where $n_{\mathrm{epoch}}$ denotes the current training epoch.

### 3.4 Training Objective

From a video-language input, we obtain $L_m$ predictions $\hat{Y} = \{(\hat{s}_i, \hat{c}_i, \hat{co}_i, \hat{w}_i)\}_{i=1}^{L_m}$. During training $L_m$ is the number of groundtruth localizations,

while during testing $L_m$ is selected based on validation. We define the matching loss $L_{\mathrm{match}}$ between predictions and groundtruth as:

$$\mathcal{L}_{\mathrm{match}} = -\frac{1}{L_m}\sum_{i=1}^{L_m}(\hat{s}_i - \lambda_1||c_i - \hat{c}_i|| -$$
$$\lambda_2||w_i - \hat{w}_i|| - \lambda_3||co_i - (\hat{co}_i - \hat{c}_i)||),$$
(30)

where $\lambda_{\{1,2,3,4\}}$ denote the hyperparameter weights for the salience, center, width, and offset losses, respectively. We jointly optimize the matching loss with the EBM negative log-likelihood (NLL) loss as follows:

$$\mathcal{L} = \mathcal{L}_{\mathrm{match}} + \lambda_{\mathrm{NLL}}\mathcal{L}_{\mathrm{NLL}},$$
(31)

where $\lambda_{\mathrm{NLL}}$ denotes the weight to scale the NLL loss size.

## 4 Experiments

### 4.1 Datasets

We evaluate our methods on four benchmark datasets for the temporal language grounding task: QVHighlights, Charades-STA, YouTube Highlights, and TVSum.

**QVHighlights** is collected by (Lei et al., 2021) to span diverse content on 3 major topics: daily vlog, travel vlog, and news. There are 10,148 videos with 18,367 moments associated with 10,310 queries. We follow (Lei et al., 2018; Liu et al., 2022) to split the dataset into 70% train, 15% val, and 15% test portions.

**Charades-STA** (Gao et al., 2017) consists of videos about daily indoor activities. The dataset is split into 12,408 and 3,720 moment annotations for training and testing, respectively.

**YouTube Highlights** is prepared by (Sun et al., 2014) to comprise six categories, i.e. dog, gymnastics, parkour, skating, skiing and surfing. In each category, we inherit the original training-testing split as benchmark for the TLG task.

**TVSum** (Hong et al., 2020) is a video summarization dataset possessing 10 event categories. We employ the video title as the language query and the training/testing split of 0.8/0.2 for experiments.

### 4.2 Experimental Settings

**Evaluation Metrics.** Our metrics include *Rank $k@\mu$*, *mAP@$\mu$*, and *Hit@1*. *Rank $k@\mu$* is the percentage of the testing samples that have at least one correct localization in the top-$k$ choices, where a localization is correct if its IoU with the groundtruth

| Method | R1@ | | mAP@ | | | HIT@1 |
|---|---|---|---|---|---|---|
| | 0.5 | 0.7 | 0.5 | 0.75 | Avg. | |
| CLIP | 16.88 | 5.19 | 18.11 | 7.00 | 7.67 | 61.04 |
| XML | 41.83 | 30.35 | 44.63 | 31.73 | 32.14 | 55.25 |
| XML+ | 46.69 | 33.46 | 47.89 | 34.67 | 34.90 | 55.06 |
| Moment-DETR | 52.89 | 33.02 | 54.82 | 29.40 | 30.73 | 55.60 |
| UMT | 56.23 | 41.18 | 53.83 | 37.01 | 36.12 | 59.99 |
| **DemaFormer** (Ours) | **62.39** | **43.94** | **58.25** | **39.36** | **38.71** | **64.77** |
| Moment-DETR w/ PT | 59.78 | 40.33 | 60.51 | 35.36 | 36.14 | 60.17 |
| UMT w/ PT | 60.83 | 43.26 | 57.33 | 39.12 | 38.08 | 62.39 |
| **DemaFormer** w/ PT | **63.55** | **45.87** | **59.32** | **41.94** | **40.67** | **65.03** |

Table 1: Temporal language grounding results on the QVHighlights dataset. "*w/ PT*" denotes pre-training with ASR captions.

| Method | R1@ | | R5@ | |
|---|---|---|---|---|
| | 0.5 | 0.7 | 0.5 | 0.7 |
| MAN | 41.24 | 20.54 | 83.21 | 51.85 |
| 2D-TAN | 39.70 | 23.31 | 80.32 | 51.26 |
| DRN | 42.90 | 23.68 | 87.80 | 54.87 |
| RaNet | 43.87 | 26.83 | 86.67 | 54.22 |
| Moment-DETR | 48.95 | 21.23 | 86.96 | 50.14 |
| UMT | 49.35 | 26.16 | 89.41 | 54.95 |
| **DemaFormer** | **52.63** | **32.15** | **91.94** | **60.13** |

Table 2: Temporal language grounding results on the Charades-STA dataset.

is larger than the threshold $\mu$. In a similar manner, $mAP@\mu$ is the mean average precision of localizations whose IoU is larger than $\mu$. *Hit@1* computes the hit ratio for the moment with the highest predicted salience score in a video. We consider a moment is hit if its groundtruth salience is larger than or equal to a threshold $\tau$. Following previous works (Lei et al., 2021; Liu et al., 2022), we adopt *Rank 1@$\mu$* with $\mu \in \{0.5, 0.75\}$ and *Hit@1* with $\tau = 4$ for the QVHighlights dataset. For the Charades-STA dataset, we use *Rank k@$\mu$* with $k \in \{1, 5\}$ and $\mu \in \{0.5, 0.75\}$. We apply *mAP* for both the TVSum and YouTube Highlights datasets. **Implementation Details.** For fair comparison with previous works (Liu et al., 2022; Lei et al., 2021), on QVHighlights, we use SlowFast (Feichtenhofer et al., 2019) and CLIP (Radford et al., 2021) to obtain features for the video moments and CLIP text encoder to obtain features for the language queries. For feature extraction of the Charades-STA dataset, we deploy VGG (Simonyan and Zisserman, 2014) and optical flow features for video moments and GloVe embeddings (Pennington et al., 2014) for language tokens. On YouTube Highlights and TVSum, we utilize the I3D model (Carreira and Zisserman, 2017) pre-trained on Kinetics 400 (Kay et al., 2017) to extract moment-level visual representations, and CLIP text encoder to extract language representations. Furthermore, as in (Liu

et al., 2022; Lei et al., 2021), for QVHighlights dataset, we also experiment with pre-training our architecture with noisy automatic speech recognition (ASR) captions before fine-tuning on the downstream training samples. For all audio features, we use the PANN model pre-trained on AudioSet (Gemmeke et al., 2017). We provide detailed hyperparameter settings in Appendix B.

### 4.3 Baselines

To evaluate the proposed methods, we compare our performance with a diversity of baselines:

- **UMT** (Liu et al., 2022): a multi-modal transformer model to handle three modalities, including audio, text, and video.
- **Moment-DETR** (Lei et al., 2021): a multi-modal transformer model that applies the original self-attention mechanism to encode no human prior and eliminates manually-designed pre-processing and post-processing procedures.
- **CLIP** (Radford et al., 2021): a framework of visual CNN and textual transformer models trained with a contrastive objective.
- **XML** (Lei et al., 2020): a framework of visual ResNet and textual RoBERTa models with a late fusion approach to fuse the visual and textual features. We include an additional variant, XML+, which is trained with the combination of our salience loss and the XML's loss.
- **RaNet** (Gao et al., 2021a): a baseline with BiLSTM text encoder, CNN image encoder, and a graph cross-modalithy interactor to learn moment-query relations and select the target moments.
- **2D-TAN** (Zhang et al., 2020b): a method to specify moment candidates as a 2D map where the row and column indices indicate the starting and ending points, respectively.
- **DRN** (Zeng et al., 2020): a dense regression network which treats all video moments as positive and seeks to predict its distance to the groundtruth starting and ending boundaries.
- **MAN** (Zhang et al., 2019): a baseline with a structured graph network to model the moment-wise temporal relationships.
- **TCG** (Ye et al., 2021): a multi-modal TLG architecture equipped with a low-rank tensor fusion mechanism and hierarchical temporal context encoding scheme.

| Method | Dog | Gym. | Par. | Ska. | Ski. | Sur. | Avg. |
|---|---|---|---|---|---|---|---|
| LIM-S | 57.90 | 41.67 | 66.96 | 57.78 | 48.57 | 65.08 | 56.38 |
| DL-VHD | 70.78 | 53.24 | 77.16 | 72.46 | 66.14 | 76.19 | 69.26 |
| MINI-Net | 58.24 | 61.68 | 70.20 | 72.18 | 58.68 | 65.06 | 64.43 |
| TCG | 55.41 | 62.69 | 70.86 | 69.11 | 60.08 | 59.79 | 63.02 |
| Joint-VA | 64.48 | 71.92 | 80.78 | 61.99 | 73.23 | 78.26 | 71.77 |
| UMT | 65.87 | 75.17 | 81.59 | 71.78 | 72.26 | 82.66 | 74.92 |
| **DemaFormer** | **71.94** | **77.61** | **86.05** | **78.92** | **74.08** | **83.82** | **77.93** |

Table 3: Temporal language grounding results on the YouTube Highlights dataset.

- **Joint-VA** (Badamdorj et al., 2021): an approach applying attention mechanism to fuse multi-modal features and a sentinel technique to discount noisy signals.
- **MINI-Net** (Hong et al., 2020): a weakly supervised learning approach that trains a positive bag of query-relevant moments to possess higher scores than negative bags of query-irrelevant moments.
- **LIM-S** (Xiong et al., 2019): a TLG approach that leverages video duration as a weak supervision signal.
- **DL-VHD** (Xu et al., 2021): a framework applying dual learners to capture cross-category concepts and video moment highlight notions.

## 4.4 Comparison with State-of-the-arts

We report results of our DemaFormer and the baselines in Table 1, 2, 3, and 4 on the QVHighlights, Charades-STA, YouTube Highlights, and TVSum datasets, respectively. As can be seen, our methods significantly outperform previous approaches.

**QVHighlights.** Compared with the previous best method UMT, our DemaFormer achieves $2\%$ absolute improvement at least across all evaluation settings of *Rank 1@$\mu$*, particularly $4.54\%$ for $\mu = 0.5$ and $2.01\%$ for $\mu = 0.7$, respectively. When pretrained with the ASR captions, our method outperforms UMT with $2.59\%$ of *mAP* on average and $2.64$ points of *Hit@1*. These results demonstrate that our method can enhance the TLG operation in diverse settings, including daily vlog, travel vlog, and news.

**Charades-STA.** We increase the performance of UMT with $3.28\%$ in terms of *Rank 1@0.5* and $2.53\%$ in terms of *Rank 5@0.5*. Upon tighter $\mu = 0.7$, we achieve a larger degree of enhancement with $5.99\%$ for *Rank 1* and $5.18\%$ for *Rank 5*. We hypothesize that this is because our energy-based modeling can focus on separating highly relevant localizations from other video moment candidates.

**TVSum.** In Table 4, we compare our model with other competitive approaches. Our architecture accomplishes the highest *mAP* scores across all categories and in overall as well. In detail, we outperform the second-best UMT up to $19.34\%$ at maximum on the BT portion. Analogous to the QVHighlights experiments, this demonstrates that our framework can better model the video-language inputs in various contexts to polish the temporal language grounding performance.

**YouTube Highlights.** Equivalent to TVSum, our DemaFormer with the energy-based modeling approach outperforms prior competitive models across various subsets. Specifically, we gain *mAP* increases of $1.16\%$ at minimum on the surfing portion and $6.07\%$ at maximum on the dog portion. We attribute such improvement to the more effective modeling operation of the proposed DEMA computation in attention, since it can exhibit local dependencies of moment-query inputs for appropriate modeling in various contexts.

## 4.5 Ablation Studies

In this section, we study the impact of (1) Damped Exponential Moving Average (DEMA), (2) Energy-Based Modeling (EBM), (3) Langevin Sampling Steps, and (4) Choice of Energy Functions.

**With vs. Without DEMA.** From Table 5, removing the damping factor results in slight performance decrease, for example $1.04\%$ and $0.26\%$ in terms of *Rank 1@0.7* on QVHighlights and Charades-STA, respectively. The main reason is that without the damping coefficient, the model lacks the ability to adjust the information injected into adjacent input elements, such that the amount could be excessive to make it hard to distinguish video moments. Moreover, we observe that completely eliminating the DEMA computation leads to significant decrease, specifically up to $2.97\%$ and $2.51\%$ of *Rank 1@0.5* respectively on QVHighlights and Charades-STA, since the model no longer specifies the moment-query distribution effectively.

**With vs Without EBM.** Investigating the last rows of Table 5, we realize that adopting energy-based modeling can improve the temporal language

| Method | VT | VU | GA | MS | PK | PR | FM | BK | BT | DS | Avg. |
|---|---|---|---|---|---|---|---|---|---|---|---|
| LIM-S | 55.92 | 42.92 | 61.21 | 53.99 | 60.36 | 47.50 | 43.24 | 66.31 | 69.14 | 62.63 | 56.32 |
| DL-VHD | 86.51 | 68.74 | 74.93 | 86.22 | 79.02 | 63.21 | 58.92 | 72.63 | 78.93 | 64.04 | 73.32 |
| MINI-Net | 80.64 | 68.31 | 78.16 | 81.83 | 78.11 | 65.82 | 57.84 | 74.99 | 80.22 | 65.49 | 73.14 |
| TCG | 84.96 | 71.43 | 81.92 | 78.61 | 80.18 | 75.52 | 71.61 | 77.34 | 78.57 | 68.13 | 76.83 |
| Joint-VA | 83.73 | 57.32 | 78.54 | 86.08 | 80.10 | 69.23 | 70.03 | 73.03 | 97.44 | 67.51 | 76.30 |
| UMT | 87.54 | 81.51 | 88.21 | 78.83 | 81.42 | 87.04 | 75.98 | 86.93 | 79.64 | 83.14 | 83.02 |
| **DemaFormer** | **88.73** | **85.15** | **92.11** | **88.82** | **82.20** | **89.18** | **80.36** | **89.06** | **98.98** | **85.24** | **87.92** |

Table 4: Temporal language grounding results on the TVSum dataset.

| Dataset | Method | R1@ 0.5 | R1@ 0.7 |
|---|---|---|---|
| QVHighlights | DemaFormer | 62.39 | 43.94 |
| | - w/o damping | 60.52 | 42.90 |
| | - w/o DEMA | 59.42 | 42.06 |
| | - w/o EBM | 59.23 | 42.32 |
| Charades-STA | DemaFormer | 52.63 | 32.15 |
| | - w/o damping | 51.05 | 31.89 |
| | - w/o DEMA | 50.12 | 30.96 |
| | - w/o EBM | 49.96 | 30.77 |

Table 5: Performance comparisons on QVHighlights and Charades-STA datasets in ablative experiments of DEMA and EBM components of DemaFormer.

| Energy Function | R1@ 0.5 | R1@ 0.7 | R5@ 0.5 | R5@ 0.7 |
|---|---|---|---|---|
| Elementwise Cosine Similarity | 51.88 | 31.08 | 90.74 | 59.59 |
| Pooling-based Cosine Similarity | 51.85 | 29.97 | 88.98 | 58.86 |
| Salience score | 52.63 | 32.15 | 91.94 | 60.13 |

Table 6: Performance comparison on the Charades-STA dataset in ablative experiments of the energy function choices.

grounding performance. Particularly, adding the EBM training objective brings enhancement of 2.67% on Charades-STA and 3.16% on QVHighlights in terms of *Rank 1@0.5*. This substantiates that the EBM can successfully capture the distribution of moment-query representations in which relevant localizations are separated from the irrelevant ones. We provide more illustrative analysis in Section 4.6 and Appendix F.

**Langevin Sampling Steps.** We investigate the impact of the number of sampling steps $K$ in our Langevin equation (24) upon DemaFormer. Figure 4 shows that DemaFormer's performance increases as $K$ grows. However, as long as $K$ passes the threshold 100, the model performance converges with negligible fluctuation. We hypothesize that at $K = 100$ the added noise is sufficient to segregate target localizations from elsewhere.

**Choice of Energy Functions.** We experiment with different choices of the energy function. Inspired by the contrastive learning works (Chuang et al., 2020; Oord et al., 2018), we compute the negative cosine similarity of video clips $O_e^v = \{\mathbf{o}_{e,i}\}_{i=1}^{L_v}$ and query tokens $O_e^q = \{\mathbf{o}_{e,i}\}_{i=L_v+1}^{L_v+L_q}$ in the ele-

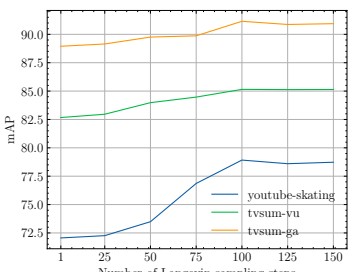

Figure 4: Effect of the number of Langevin sampling steps upon localization performance on the VU and GA portions of the TVSum dataset and the skating portion of the YouTube Highlights dataset..

Text query: The woman wearing sunglasses crosses a small colorful bridge over the river

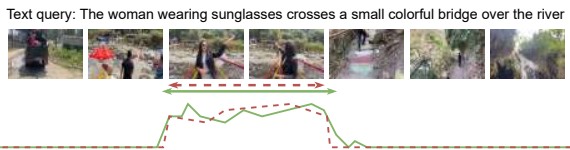

Figure 5: Qualitative visualization of DemaFormer model. Green arrow line denotes the predicted localization and green normal line the predicted salience scores. Red arrow line denotes the groundtruth localization and red normal line the annotated salience scores.

mentwise and pooling-based manner (we provide the formulations in Appendix E), and evaluate the performance of the variants in Table 6. As the comparison shows, directly utilizing the salience score provides localizations with the most accuracy. This suggests that similarity functions do not fully implement the query-relevance concept.

### 4.6 Qualitative Analysis

We illustrate a prediction example from the QVHighlights dataset by our DemaFormer in Figure 1, 2 and 5. We observe that our model correctly localizes target moments with respect to the user query. Our predicted salience scores also align with the groundtruth scores, which are measured by averaging the three annotated scores in the dataset. In addition, we also utilize t-SNE to visualize the moment-query representations of the example in Figure 1. We realize that the representations of the target localizations stay separately from the remaining ones, whereas those from the UMT model do mingle together. This could explain the accurate localilzation of DemaFormer and verifies the effec-

tive modeling of the proposed DEMA mechanism combined with the energy-based modeling. We provide more examples in Appendix F.

## 5 Conclusion

In this paper, we propose DemaFormer, a novel neural architecture for the temporal language grounding (TLG) task. By leveraging the exponential moving average approach with a damping factor, DemaFormer is capable of incorporating local dependencies among moment-query localizations. Additionally, we propose an energy-based strategy to explicitly model localization distribution. On four public benchmarks for the TLG task, our method is effective and outperforms state-of-the-art approaches with a significant margin.

## 6 Limitations

Our framework requires negative sampling via the Langevin dynamics equation. This incurs additional compute cost while training the language grounding model. Also, although we propose general methods to enhance the grounding performance, we have not studied their impact in cross-domain scenarios, where the model is trained upon one domain (e.g. skiing videos) and tested upon another (e.g. skating videos). We leave these gaps as future work to optimize our framework in more diverse contexts and use cases.

## 7 Acknowledgement

This research/project is supported by the National Research Foundation, Singapore under its AI Singapore Programme (AISG Award No: AISG3-PhD-2023-08-051T). Thong Nguyen is supported by a Google Ph.D. Fellowship in Natural Language Processing.

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

| Losses | R1@ | | mAP@ | | | HIT@1 |
| --- | --- | --- | --- | --- | --- | --- |
| | 0.5 | 0.7 | 0.5 | 0.75 | Avg. | |
| $\mathcal{L}_s + \mathcal{L}_c + \mathcal{L}_w$ | 57.29 | 40.97 | 54.29 | 36.61 | 35.98 | 60.84 |
| $\mathcal{L}_s + \mathcal{L}_c + \mathcal{L}_w + \mathcal{L}_{oc}$ | 62.39 | 43.94 | 58.25 | 39.36 | 38.71 | 64.77 |

Table 7: Performance comparison on the QVHighlights dataset in ablative experiments of the influence of localization loss terms on our DemaFormer model.

## A    Gradient of the Energy-Based Negative Log-Likelihood Objective

We have the specification of the distribution of the moment-query representations:

$$p_\theta(\mathbf{o}) = \frac{\exp(-E_\theta(\mathbf{o}))}{Z_\theta}, \tag{32}$$

$$Z_\theta = \int \exp(-E_\theta(\mathbf{o}))\mathrm{d}\mathbf{o}. \tag{33}$$

We differentiate the negative log-likelihood of the representation:

$$-\frac{\delta \log(p_\theta(\mathbf{o}))}{\delta \theta} \tag{34}$$

$$= -\frac{\delta}{\delta \theta} \log\left(\frac{1}{Z_\theta} \exp(-E_\theta(\mathbf{o}))\right) \tag{35}$$

$$= -\frac{\delta}{\delta \theta} \left(-\log Z_\theta - E_\theta(\mathbf{o})\right) \tag{36}$$

$$= -\frac{\delta}{\delta \theta} \left(-\log \int \exp(-E_\theta(\mathbf{o}))\mathrm{d}\mathbf{o} - E_\theta(\mathbf{o})\right) \tag{37}$$

$$= -\frac{1}{Z_\theta} \left(\int \exp(-E_\theta)\frac{\delta E_\theta}{\delta \theta}\mathrm{d}\mathbf{o}\right) + \frac{\delta E_\theta(\mathbf{o})}{\delta \theta} \tag{38}$$

$$= -\left(\frac{\delta E_\theta(\mathbf{o}')}{\delta \theta}\right)_{\mathbf{o}' \sim p_\theta} + \frac{\delta E_\theta(\mathbf{o})}{\delta \theta} \tag{39}$$

$$= \nabla_\theta E_\theta(\mathbf{o}^+) - \nabla_\theta E_\theta(\mathbf{o}^-). \tag{40}$$

We obtain the gradient formulation for the energy-based modeling component.

## B    Hyperparameter Settings

Our hidden dimension for the key tensor is $d_K = 256$. Regarding our energy-based models (EBM), we adapt $K = 100$ and $\gamma = 0.1$ for all datasets. For the training procedure, we adopt $\lambda_1 = \frac{1}{3}, \lambda_2 = 0.01, \lambda_3 = \frac{1}{3}, \lambda_{\text{NLL}} = 0.1$, and $\alpha_{\min} = 0.1$. To extract positive samples to train the EBMs, we use $\rho = 4, 1, 1$, and $0.4$ for the QVHighlights, Charades-STA, YouTube Highlights, and TVSum datasets, respectively. During testing, we adopt $L_m = 10$ based on the validation performance. The number of DemaFormer encoder and decoder layers are set as $N_e = N_d = 2$. For all training settings, we utilize the Adam optimizer with learning rate $1e - 3$ and weight decay $1e - 4$. We train our DemaFormer model with batch size 32 for 200 epochs on the QVHighlights, batch size 8 for 100 epochs on the Charades-STA, batch size 4 for 100 epochs on the YouTube Highlights, and batch size 1 for 500 epochs on the TVSum dataset, respectively.

## C    Localization Losses

Following (Liu et al., 2022; Lei et al., 2021), we conduct experiments to study the effect of localization losses on our DemaFormer architecture. We define $\mathcal{L}_s = -\frac{1}{L_m} \sum_{i=1}^{L_m} \hat{s}_i$ to be the salience loss term,

| Activation Function | R1@ | | mAP@ | | | HIT@1 |
|---|---|---|---|---|---|---|
| | 0.5 | 0.7 | 0.5 | 0.75 | Avg. | |
| Tanh | 62.38 | 43.93 | 58.23 | 39.32 | 38.70 | 64.73 |
| ReLU | 62.36 | 43.90 | 58.22 | 39.27 | 38.68 | 64.71 |
| GELU | 62.34 | 43.89 | 58.18 | 39.26 | 38.66 | 64.67 |
| SiLU | 62.39 | 43.94 | 58.25 | 39.36 | 38.71 | 64.77 |

Table 8: Performance comparison on the QVHighlights dataset in ablative experiments of the activation functions.

$\mathcal{L}_c = \frac{1}{L_m} \sum_{i=1}^{L_m} ||c_i - \hat{c}_i||$ to be the center loss term, $\mathcal{L}_w = \frac{1}{L_m} \sum_{i=1}^{L_m} ||w_i - \hat{w}_i||$ the width loss term, and $\mathcal{L}_{co} = \frac{1}{L_m} \sum_{i=1}^{L_m} ||co_i - \hat{co}_i||$ the center offset loss term. Because the salience, center and width terms are mandatory, we justify the necessity of the center offset term. As can be seen from Table 7, with the center offset term the localization scores increase from $54.29\%$ to $58.25\%$ of *mAP@0.5*, and from $40.97\%$ to $43.94\%$ of *Rank 1@0.7*. This demonstrates that the center offset term helps our DemaFormer architecture predict the localizations more precisely.

## D   Choice of Activation Functions

In this appendix, we adopt different activation functions for our DEMA attention in Section 3.1 and compare their performances. In detail, we experiment with the Tanh (Dubey et al., 2022), ReLU (Dubey et al., 2022), and GELU functions (Hendrycks and Gimpel, 2016). We report the temporal language grounding performance with these activation functions on the QVHighlights dataset in Table 8. We observe that DemaFormer exhibits negligible performance fluctuation. These results demonstrate the robustness of our proposed DemaFormer with respect to the choice of activation functions.

## E   Specification of Energy Functions

We provide the formulation of energy functions we experiment with in Table 6.

- Element-wise cosine similarity:

$$E_\theta(\mathbf{o}_{d,i}) = -\frac{1}{L_q} \sum_{j=L_v+1}^{L_v+L_q} \frac{\mathbf{o}_{e,i} \cdot \mathbf{o}_{e,j}}{|\mathbf{o}_{e,i}| \cdot |\mathbf{o}_{e,j}|}. \quad (41)$$

- Pooling-based cosine similarity:

$$\mathbf{q} = \text{MaxPool}\{\mathbf{o}_{e,j}\}_{j=L_v+1}^{L_v+L_q}, \quad (42)$$

$$E_\theta(\mathbf{o}_{d,i}) = -\frac{\mathbf{o}_{e,i} \cdot \mathbf{q}}{|\mathbf{o}_{e,i}| \cdot |\mathbf{q}|}. \quad (43)$$

- Salience score:

$$E_\theta(\mathbf{o}_{d,i}) = -\hat{s}_i. \quad (44)$$

## F   More prediction examples

In this appendix, we present more predictions of our DemaFormer model in Figure 6, 7, and 8.

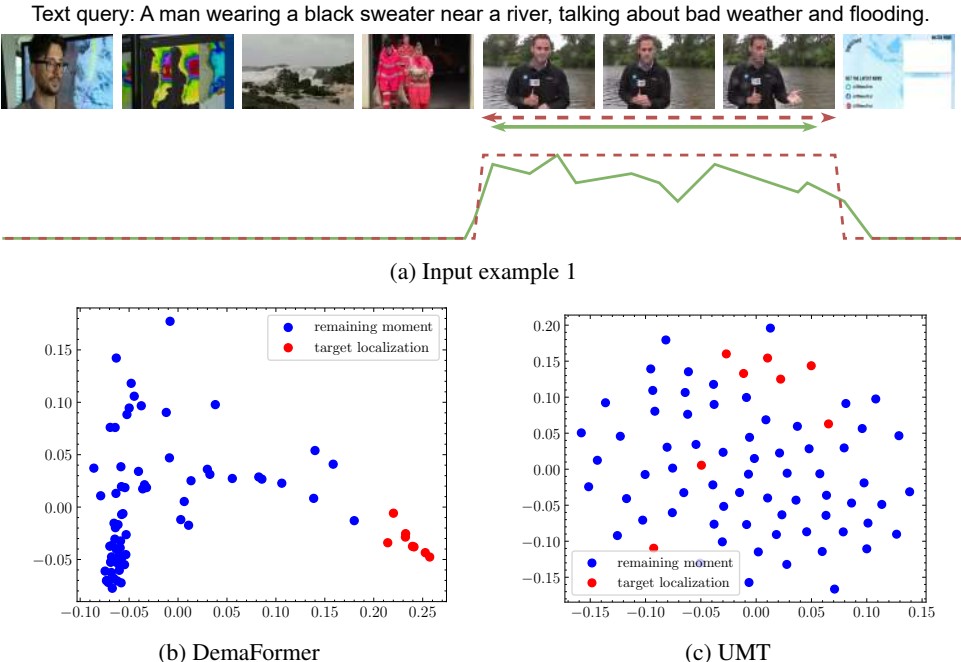

Figure 6: Prediction example 1 with the t-SNE visualizations of the DemaFormer model and the UMT model. Green arrow line denotes the predicted localization and green normal line the predicted salience scores. Red arrow line denotes the groundtruth localization and red normal line the annotated salience scores.

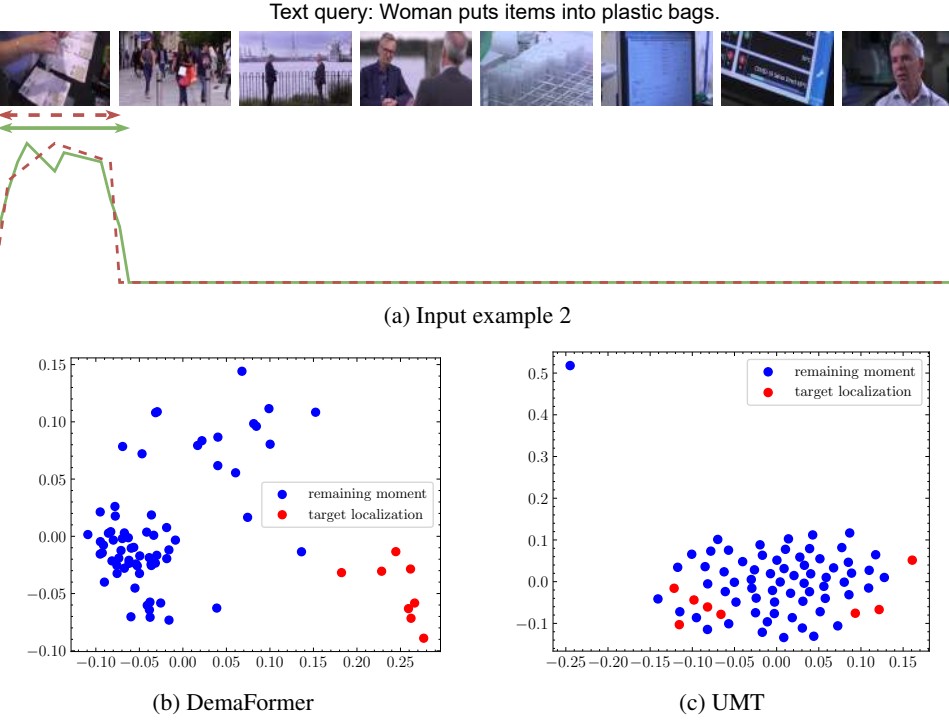

Figure 7: Prediction example 2 with the t-SNE visualizations of the DemaFormer model and the UMT model. Green arrow line denotes the predicted localization and green normal line the predicted salience scores. Red arrow line denotes the groundtruth localization and red normal line the annotated salience scores.

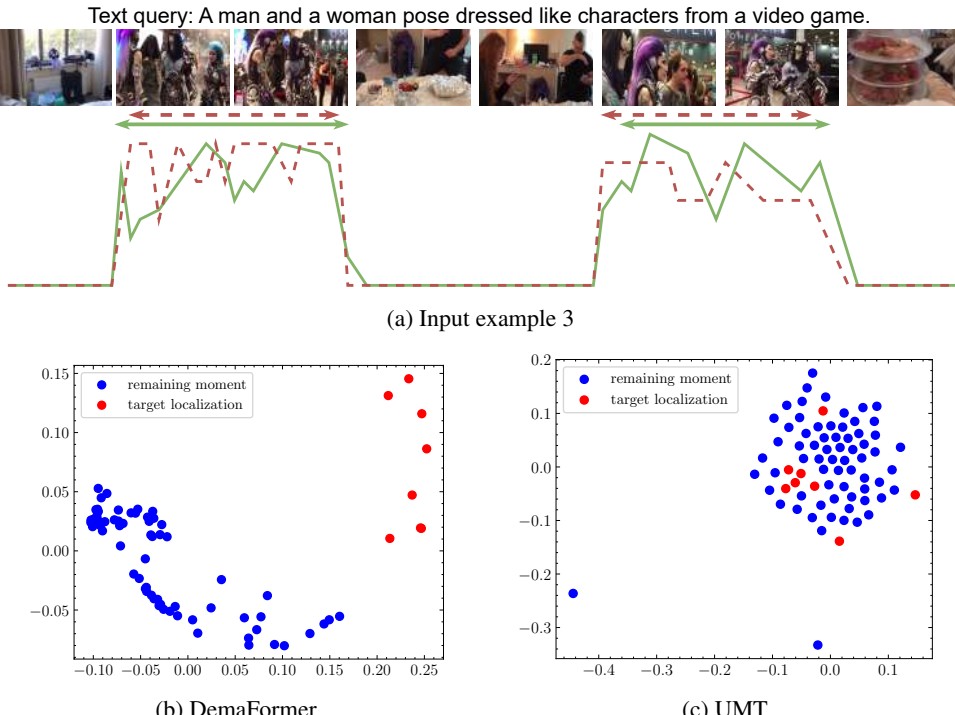

(a) Input example 3

(b) DemaFormer

(c) UMT

Figure 8: Prediction example 3 with the t-SNE visualizations of the DemaFormer model and the UMT model. Green arrow line denotes the predicted localization and green normal line the predicted salience scores. Red arrow line denotes the groundtruth localization and red normal line the annotated salience scores.