# OpenReview forum: "DemaFormer: Damped Exponential Moving Average Transformer with Energy-Based Modeling for Temporal Language Grounding"
_EMNLP/2023/Conference — EMNLP 2023 Findings_

### Official Review · Reviewer_SbrT · 2023-07-23

**Soundness:** 2

**Excitement:**

3: Ambivalent: It has merits (e.g., it reports state-of-the-art results, the idea is nice), but there are key weaknesses (e.g., it describes incremental work), and it can significantly benefit from another round of revision. However, I won't object to accepting it if my co-reviewers champion it.

**Paper Topic And Main Contributions:**

Dear Area Chair,

I am not an expert in this field. If possible, could you please assign this paper to other reviewers? Thanks!

**Reasons To Accept:**

N/A

**Reasons To Reject:**

N/A

**Reproducibility:**

3: Could reproduce the results with some difficulty. The settings of parameters are underspecified or subjectively determined; the training/evaluation data are not widely available.

**Reviewer Confidence:**

1: Not my area, or paper was hard for me to understand. My evaluation is just an educated guess.

---

> ### Author Rebuttal · Authors · 2023-08-28
>
> Thank you for your sincere note. If you have chances to review our future paper, we look forward to your insightful comments and feedback.

---

### Official Review · Reviewer_ecVy · 2023-08-04

**Typos Grammar Style And Presentation Improvements:** None.
**Soundness:** 4

**Excitement:**

4: Strong: This paper deepens the understanding of some phenomenon or lowers the barriers to an existing research direction.

**Missing References:**

[1] Learning energy-based models by diffusion recovery likelihood, ICLR 2021.

[2] Vaebm: A symbiosis between variational autoencoders and energy-based models, ICLR 2021.

**Paper Topic And Main Contributions:**

In the Temporal Language Grounding task, naive attention is used in existing research to learn the relationship between video moments and textual queries, which results in ineffective distributions between features. To address this problem, this paper uses an energy-based model to achieve contrasting distributions of video representations. In addition, DemaFormer is proposed that uses the exponential moving average with a learnable damping factor to effectively encode moment-query inputs.

**Questions For The Authors:**

1.	In Line 77-79, the authors proposed to use an unsupervised training model. How is it unsupervised?
2.	Unlike Charades-STA, in QVHighlighs, a query text will correspond to multiple video clips, can the learned moment-Query Representations handle multiple video clips? This paper seems only describe one target video clip case.
3.	The author wrote in Line 313 that as training proceeds, the negative samples get closer to the positive samples, does this mean that the representations are gradually similar? What is the criterion for identifying the positive and negative samples during model training?

**Reasons To Accept:**

1.	This paper takes a novel perspective, i.e. introducing EBM, which distinguishes between target moments and other moments in the video by maximizing and minimizing the energy.
2.	The designed DEMA module seems to be novel.
3.	The state-of-the-art results are achieved on the four public datasets.

**Reasons To Reject:**

1.	There is a lack of the motivation to introduce EBM. It is not clear to me why this module is helpful to model the distribution of moment-query representations. Besides, what is the meaning of energy here?
2.	It is also not clear to me why the adjacent video features are helpful to model joint moment-query representation (Line 64-68). In my opinion, the temporally related video features should contain much redundancy, which not only increases computation cost, but also introduces noises.
3.	There is no description of EBM related works in Sec. 2 Related Work. The authors might consider introducing some of the classic methods [1-2].

[1] Learning energy-based models by diffusion recovery likelihood, ICLR 2021.

[2] Vaebm: A symbiosis between variational autoencoders and energy-based models, ICLR 2021.

**Reproducibility:**

4: Could mostly reproduce the results, but there may be some variation because of sample variance or minor variations in their interpretation of the protocol or method.

**Reviewer Confidence:**

4: Quite sure. I tried to check the important points carefully. It's unlikely, though conceivable, that I missed something that should affect my ratings.

---

> ### Author Rebuttal · Authors · 2023-08-28
>
> We thank the reviewer for helpful comments. We appreciate your perspective and evaluation, and we are committed to polishing our work. We hope that our response can resolve your concerns and improve your ratings.
>
> ===========================================================================================
>
> **CLARIFICATION**
>
> **_1. The motivation to introduce EBM._**
>
> The motivation to introduce EBM is to model the distribution of target video representations (lines 79-85). By maximizing this distribution, we are able to have better separable representations that are useful for the grounding task (figure 1 and 2). In contrast, traditional Transformers fail to differentiate target video moments from the others well (as in lines 71-73; figure 1 left). It is possible to substitute EBM with other distribution modeling techniques. The reason why we choose EBM lies in that EBM provides a principled and flexible way for distribution modeling and likelihood maximization, and the negative energy score can be used as the saliency score in our model (lines 85-92; Eq. 27 on line 323).
>
> **_What is the meaning of energy here?_**
>
> In the original EBM [3, 10], lower energy implies higher likelihood. In the proposed method, we additionally consider the negative energy score as a surrogate of the moment-query salience scores (in Eq. 27). This helps our learning of moment-query representations benefit from the modeling of the distribution of target video representations better.
>
> ------------------------------------
>
> **_2. Why are the adjacent video features helpful to model joint moment–query representation (Line 64-68)? The temporally related video features should contain much redundancy._**
>
> We agree that temporally neighboring frames should have redundant information. However, they also contain information that is crucial for each other. E.g., a previous frame may contain important context that is missing in the current frame, and borrowing this context information from adjacent frames to the current frame would facilitate better predictions. This is precisely why we propose the exponential moving average to capture such dependencies. As our weighting coefficient and damping coefficient are learnable during training (as in Eq 2 and line 202-203), it can be considered that we automatically learn the best trade-off between keeping useful information and abandoning redundant information from adjacent frames.
>
> We thank the reviewer for the insightful question and have added this discussion to the paper to avoid further confusion.
>
> ------------------------------------
>
> **_3. There is no description of EBM related works in Sec. 2 Related Work._**
>
> Thank you for the valuable comments. [1] is about using EBM to learn the image distribution conditioned on another distribution generated by diffusion model, while [2] is about using EBM to resolve the VAE’s problem of generating out-of-distribution samples, all of which provide informative perspectives into our work. We thank the reviewer again and we have added a discussion about [1] and [2] to the related work section as you suggested.
>
> ===========================================================================================
>
> **ANSWERS TO QUESTIONS**
>
> **_1. In Line 77-79, the authors proposed to use an unsupervised training model. How is it unsupervised?_**
>
> Our final objective is a linear composition of $L_{\text{match}}$ and $L_{\text{NLL}}$, where $L_{\text{match}}$ is supervised and $L_{\text{NLL}}$ is unsupervised in the sense of that $L_{\text{NLL}}$ aims to maximize the likelihood of video moment inputs.
>
> ------------------------------------
>
> **_2. Can the learned moment-query representations handle multiple video clips?_**
>
> Yes, our learned moment-query representations can handle multiple video clips. During inference, we forward representations of multiple clip candidates to the prediction head and only select those whose predicted salience scores are larger than a threshold. We thank the reviewer for pointing out this confusion and we have added a sentence to the paper to avoid any further confusion.
>
> ------------------------------------
>
> **_3. The author wrote in Line 313 that as training proceeds, the negative samples get closer to the positive samples, does this mean that the representations are gradually similar?_**
>
> Yes, you are right. With the gradual convergence of negative samples towards positive ones, our EBM will learn to capture the underlying patterns of target moment-query localizations better. Such phenomenon is similar to the “hard negatives” discussed in [4,5].
>
> **_What is the criterion for identifying the positive and negative samples during model training?_**
>
> Following [6], we explicitly extract positive samples from the data and draw negative ones from the probabilistic model, i.e. our EBM (as mentioned in Eq. 24, line 297). By doing so, we are able to learn better moment-query representations as shown in Figure 1.
>
> ===========================================================================================
>
> **References:**
>
> [1] Learning energy-based models by diffusion recovery likelihood, ICLR 2021.
>
> [2] Vaebm: A symbiosis between variational autoencoders and energy-based models, ICLR 2021.
>
> [3] Your classifier is secretly an energy based model and you should treat it like one, ICLR 2020.
>
> [4] Contrastive Learning with Hard Negative Samples, ICLR 2020.
>
> [5] Hard Negative Mixing for Contrastive Learning, NeurIPS 2020.
>
> [6] Generative adversarial nets, NeurIPS 2014.
>
> [7] Instance-Conditioned GAN, NeurIPS 2021.
>
> [8] A Progressive Conditional Generative Adversarial Network for Generating Dense and Colored 3D Point Clouds, International Conference on 3D Vision (3DV) 2020.
>
> [9] A survey on generative adversarial networks: Variants, applications, and training, ACM Computing Surveys 2021.
>
> [10] How to Train Your Energy-Based Models, arXiv 2021.

---

### Official Review · Reviewer_goDS · 2023-08-09

**Soundness:** 3

**Excitement:**

3: Ambivalent: It has merits (e.g., it reports state-of-the-art results, the idea is nice), but there are key weaknesses (e.g., it describes incremental work), and it can significantly benefit from another round of revision. However, I won't object to accepting it if my co-reviewers champion it.

**Paper Topic And Main Contributions:**

This paper proposes a framework that helps match video frames and langauge.  The framework contains a moving-average mechanism that associate current time with previous time, a contrastive objective which aims to increase the divergence between the unrelated video and langauge and reduces the related ones . The framework achieves improvements over baselines on many temporal langauge grounding tasks.

**Questions For The Authors:**

A. Why does the contrastive objective works better than the suprvised objective?

B. You only do moving average on representaitons,  DEMA Attention may not be approriate.  Why do we need moving average, since self-attention may have captured global representaitons.

**Reasons To Accept:**

1. This paper finds that implicit relation learning is ineffective, and proposes a effective solution to  adress the problem in expriments.

2.  This paper is relatively complete, it has strong motivations, well-conducted model and training design, and satisfied experiments.

**Reasons To Reject:**

The use of moving average and contrastive objective lacks understandings.  Refer to Questions for details.



**Reproducibility:**

3: Could reproduce the results with some difficulty. The settings of parameters are underspecified or subjectively determined; the training/evaluation data are not widely available.

**Reviewer Confidence:**

3: Pretty sure, but there's a chance I missed something. Although I have a good feel for this area in general, I did not carefully check the paper's details, e.g., the math, experimental design, or novelty.

---

> ### Author Rebuttal · Authors · 2023-08-28
>
> Thank you for your insightful comments and feedback. We are happy to listen to your perspective and dedicated to polishing our work. We hope our response is able to address your questions.
>
> ===========================================================================================
>
> **A. Why does the contrastive objective work better than the supervised objective?**
>
> Our final objective is a linear composition of $L_{\text{match}}$ and $L_{\text{NLL}}$ (Eq. 31, line 342), where $L_{\text{match}}$ is a supervised loss that directly encourages grounding and $L_{\text{NLL}}$ is a loss that maximizes the likelihood of “correct” video-text joint representations. As for the contrastive loss (as in Eq. 26), it is not to substitute the supervised $L_{\text{match}}$ loss. Rather, the contrastive loss serves as a tool to make the maximization of $L_{\text{NLL}}$ possible.
>
> **We do not want to claim that the contrastive objective works better than the supervised objective.** Rather, we believe that  $L_{\text{match}}$ and $L_{\text{NLL}}$ work together towards better grounding performance. If we only use the supervised $L_{\text{match}}$ objective, we will be less able to distinguish the target moments from the remaining ones (figure 1), and thus achieve less satisfying predictions (figure 2; section 4.4). We thank the reviewer for pointing out the confusion and have added this discussion to the paper.
>
> ===========================================================================================
>
> **B. Why do we need moving average, since self-attention may have captured global representations?**
>
> In our framework, for each video frame we get a representation that is used to predict the center and the moment width, which is used as the prediction of grounding. Therefore, compared to global representations, we are more concerned with **local dependency** between temporally adjacent frames (as mentioned in lines 63-67).
>
> As for self-attention, we agree that a self-attention layer has enough modeling capability and thus has the potential to capture local dependency. However, as it initially aims to model all the pair-wise relations, it necessitates a lot of high-quality data to capture the desired dependency, as mentioned in [1, 2]. In contrast, the proposed exponential moving average in itself can well capture the local dependency and thus serves as a kind of prior knowledge / inductive bias for better grounding performance.
>
> We thank the reviewer for the question and have added this discussion to the paper to avoid further confusion.
>
> ===========================================================================================
>
> **References:**
>
> [1] Co-Advise: Cross Inductive Bias Distillation, CVPR 2022.
>
> [2] How Do Vision Transformers Work?, ICLR 2022.

---

### Official Review · Reviewer_ZP2j · 2023-08-10

**Soundness:** 3

**Excitement:**

2: Mediocre: This paper makes marginal contributions (vs non-contemporaneous work), so I would rather not see it in the conference.

**Paper Topic And Main Contributions:**

This paper proposes DemaFormer, a novel architecture for temporal language grounding. Since naive attention might not be able to appropriately capture the relations between video moments and  the text query, this paper introduces an energy-based model framework to explicitly learn moment-query distributions. A Transformer-based architecture utilizes exponential moving average with a learnable damping factor to effectively encode moment-query inputs.

**Reasons To Accept:**

The proposed method sounds reasonable although I do not check the reproducibility of provided codes carefully.
Convincing ablation studies.
The motivation of this paper is clear and easy to understand.

**Reasons To Reject:**

The performance is not state-of-the-art, e.g., in [1], on the Charades-STA dataset, its results are 68.98 (R1@0.5), 47.79 (R1@0.7), 96.82 (R5@0.5), 75.41 (R5@0.7). However, your results are 52.63 (R1@0.5), 32.15 (R1@0.7), 91.94 (R5@0.5), 60.13 (R5@0.7). You and [1] use optical flow features. Why donot you cite and compare it?

Missing some details. e.g., how to obtain \lambda_{NLL}? No parameter analysis experiments.

Some typos, e.g., in line 385, it should be \mu \in {0.5, 0.7} not {0.5, 0.75}. In Eq. (1), the comma should be replaced with period.

[1] Liu, Daizong, et al. "Exploring optical-flow-guided motion and detection-based appearance for temporal sentence grounding." IEEE Transactions on Multimedia (2023).

**Reproducibility:**

4: Could mostly reproduce the results, but there may be some variation because of sample variance or minor variations in their interpretation of the protocol or method.

**Reviewer Confidence:**

5: Positive that my evaluation is correct. I read the paper very carefully and I am very familiar with related work.

---

> ### Author Rebuttal · Authors · 2023-08-28
>
> We thank the reviewer for the insightful comments. We genuinely value your expertise and perspective, and we are committed to enhancing our work. We hope that our response can address your concerns.
>
> ===========================================================================================
>
> **1. You and [1] use optical flow features. Why do not you cite and compare it?**
>
> Thank you for mentioning this work. We respectfully note that [1] has multiple computational-heavy components including C3D, Faster-RCNN, and optical flow, and takes advantage of the use of extra video features, while the proposed method and all other baselines only use optical flow features. Therefore, directly comparing [1] with the proposed method and other baselines is not fair. For a fair comparison we follow the experimental setting of [2,3] where [1] is not included.
>
> We agree that [1] should be cited and it would be even better if we can have an extra comparison where the proposed method and baselines all employ the same extra features as [1]. However, the authors of [1] have not released their code yet (though we have emailed them already), and thus we are not able to conduct such a comparison.
>
> We thank the reviewer again for the comment and will cite [1] together with adding the above discussion to our paper to avoid further confusion.
>
> ===========================================================================================
>
> **2. Missing some details about how to obtain $\lambda_{\text{NLL}}$?**
>
> $\lambda_{\text{NLL}}$ is a hyper-parameter to be tuned. In our implementation, we choose $\lambda_{\text{NLL}} = 0.1$ (mentioned in Appendix B). This is to ensure that the scale of $L_{\text{NLL}}$ is approximately equal to $L_{\text{match}}$ and empirically this choice of $\lambda_{\text{NLL}}$ works well. We have added this discussion to the paper to avoid further confusion.
>
> ===========================================================================================
>
> **3. Parameter analysis experiments?**
>
> We respectfully note that we do have conducted and included analysis of the choices of important parameters in our paper. Specifically, we conducted analysis upon the Langevin sampling steps (section 4.5, figure 4), the choice of energy functions (section 4.5, table 6), and the activation function of our DEMA attention (appendix D, table 8). Do you have any additional parameters that you expect us to analyze? Any further suggestions would be appreciated.
>
> ===========================================================================================
>
> **4. Some typos, e.g., in line 385, it should be $\mu \in \\{0.5, 0.7\\}$ not $\\{0.5, 0.75\\}$. In Eq. (1), the comma should be replaced with period.**
>
> We thank the reviewer for pointing them out and have revised accordingly.
>
> ===========================================================================================
>
> **References:**
>
> [1] Exploring optical-flow-guided motion and detection-based appearance for temporal sentence grounding, IEEE Transactions on Multimedia 2023.
>
> [2] UniVTG: Towards Unified Video-Language Temporal Grounding, ICCV 2023.
>
> [3] Deco: Decomposition and reconstruction for compositional temporal grounding via coarse-to-fine contrastive ranking, CVPR 2023.

---

### Meta-Review · Area_Chair_PKsP · 2023-09-18

**Recommendation:** 3

**Metareview:**

This paper introduces DemaFormer, a novel architecture designed for temporal language grounding. Since simple attention mechanisms may not effectively capture the relationships between video segments and textual queries, this paper introduces an energy-based model framework to explicitly learn distributions between moments and queries. The Transformer-based architecture employs an exponential moving average with a trainable damping factor to efficiently encode input from moments and queries. In summary, the proposed method is reasonable with clear and strong motivation. The experiments are well-conducted. In rebuttal period, the author addressed several concerns from reviewers. In particular, authors have performed an extra comparison where the proposed method and baselines all employ the same extra features as [1]. Although there was no clear answer from the reviewers, we can conclude that the results obtained from the experiment largely dispelled the reviewers' concerns about the performance. The clarification and discussion in the rebuttal should be added to the revised version. This paper has a high possibility of being accepted to findings.

---

### Decision · Program_Chairs · 2023-10-07

**Decision:**

Accept-Findings

**Comment:**

This paper introduces DemaFormer, a novel architecture designed for temporal language grounding. Since simple attention mechanisms may not effectively capture the relationships between video segments and textual queries, this paper introduces an energy-based model framework to explicitly learn distributions between moments and queries. The Transformer-based architecture employs an exponential moving average with a trainable damping factor to efficiently encode input from moments and queries. In summary, the proposed method is reasonable with clear and strong motivation. The experiments are well-conducted. In rebuttal period, the author addressed several concerns from reviewers. In particular, authors have performed an extra comparison where the proposed method and baselines all employ the same extra features as [1]. Although there was no clear answer from the reviewers, we can conclude that the results obtained from the experiment largely dispelled the reviewers' concerns about the performance. The clarification and discussion in the rebuttal should be added to the revised version. This paper has a high possibility of being accepted to findings.